# Cross-Server Computation Offloading for Multi-Task Mobile Edge Computing

**Yongpeng Shi \*** **, Yujie Xia and Ya Gao**

Henan Key Laboratory of E-Commerce Big Data Processing and Analysis, School of Physics and Electronic Information, Luoyang Normal University, Luoyang 471934, China; yjxia_2001@163.com (Y.X.); gaoya@lynu.edu.cn (Y.G.)
*   Correspondence: syp@lynu.edu.cn

**Abstract:** As an emerging network architecture and technology, mobile edge computing (MEC) can alleviate the tension between the computation-intensive applications and the resource-constrained mobile devices. However, most available studies on computation offloading in MEC assume that the edge severs host various applications and can cope with all kinds of computation tasks, ignoring limited computing resources and storage capacities of the MEC architecture. To make full use of the available resources deployed on the edge servers, in this paper, we study the cross-server computation offloading problem to realize the collaboration among multiple edge servers for multi-task mobile edge computing, and propose a greedy approximation algorithm as our solution to minimize the overall consumed energy. Numerical results validate that our proposed method can not only give near-optimal solutions with much higher computational efficiency, but also scale well with the growing number of mobile devices and tasks.

**Keywords:** mobile edge computing; cross-server computation offloading; energy consumption minimization; greedy algorithm

## 1. Introduction

The past decade has witnessed wireless communications experiencing an explosive growth in terms of both the number of mobile devices (MDs) and services to be supported [1]. Lots of new mobile applications (APPs) like interactive gaming, natural language processing, and face identification have emerged and attracted great attention [2]. These kinds of APPs usually require rich computation resource to process their large amount of data. However, due to the limited physical size, the battery-powered and resource-constrained MDs cannot meet their Quality of Service (QoS) requirements of low computing latency and energy consumption. The conflict between resource-hungry APPs and resource-constrained MDs poses an unexampled challenge to deploy the coming new generation mobile networks and Internet of Things (IoT).

As a promising technology to cope with the ever-rising computation demands, mobile edge computing (MEC) [3,4] can alleviate the tension between the computation-intensive APPs and the resource-constrained MDs. MEC integrates cloud computing functionalities into mobile systems efficiently by deploying MEC servers (MECSs) at the edge of pervasive wireless access networks. Unlike conventional cloud computing systems which needs long propagation delay for data transmission, MEC can offer powerful computation capability in close proximity to MDs. Offloading the MD's computation tasks to the MECS can significantly improve the computation efficiency including processing latency and energy consumption [5].

Recently, plenty of ink has been poured on the problem of mobile edge computation offloading (MECO) [6,7], and many novel offloading schemes have been proposed to optimize either computing

overhead [8,9] or communication resources [10,11]. From the aspects of mobile user number, type of computation tasks, and the involved MEC servers, these research work mainly considered the network scenarios of multi-user single server [12], multi-user multi-server [13], multi-task multi-server [14], and multi-user multi-task multi-server [15,16]. There is no doubt that the related literature provides precious viewpoints for the performance optimization and resource allocation of MEC. However, most existing MECO studies usually assume the MEC severs host various applications and can handle all the MDs' computation tasks. In fact, constrained by the high cost of infrastructure deployment and maintenance, the computing and storage capacities of a MECS cannot be as large as unbounded. Therefore, it is impractical for one MECS to be deployed all types of applications to provide computing services for multitudinous tasks. How to use the limited number of APP types hosted on MECSs to devise effective multi-server collaborative task offloading strategies, is deserving further investigation.

Toward this end, we present this paper to discuss the cross-server computation offloading problem for multiple tasks with multi-server in MEC networks, where multiple MDs request various types of computation tasks, and the MECSs are deployed limited computing capabilities. A greedy cross-server offloading solution is proposed to optimize the energy consumption with the constraints of accomplishing latency and computing resources. In particular, the main contributions of this paper are summarized as follows.

- Given the multi-user multi-server multi-task mobile edge computing network architecture, we mainly study the problem of cross-server computation offloading, which considers how to improve the utility of the limited computation resources deployed on edge servers in MEC.
- We first formally formulate the cross-server multiple task computation offloading problem to optimize the total energy consumption given the constraints of task accomplishing time and the computing resources hosted on the MECSs. Then a greedy energy-aware task offloading algorithm, i.e., GAA, is presented to solve this problem. Compared to the basic exhaustive algorithm (BEA), GAA can obtain the approximate optimal consumed energy with computational complexity of $O(n \cdot m^2)$, which is much more efficient than BEA with running time of $O(3^n)$. Here, $n$ and $m$ denote the number of tasks and MECSs, respectively.
- Extensive experiments have been performed to verify the efficiencies of our proposed algorithms. Performance evaluation shows that for both different number of MDs and various computing models, GAA can always give the optimal consumed energy very close to BEA, while taking much short running time.

The remainder of this paper is structured as follows: Section 2 introduces the related work in recent years. Section 3 presents the system model. In Section 4 we first define the problem of cross-server multi-task computation offloading, then propose two detailed algorithms, BEA and GAA. Section 5 presents the extensive performance evaluation, and finally Section 6 concludes the whole paper.

## 2. Related Work

Over recent years, the computation offloading for MEC has attracted much attention from academia to industry, and plenty of research has been carried out on such a challenging issue [17,18]. Among them, most research studied the problem of offloading multi-user's tasks to one single MECS. For instance, Qin et al. introduced a distributed non-cooperative game model for multiple users media MECO in MEC to achieve the maximum benefits in terms of transmission time, energy cost, and computation cost [19]. Zheng et al. [20] presented a distributed multi-agent stochastic learning algorithm to reduce computation cost for the multi-user MECO problem. In [21], to jointly optimize the overall completing delay and the users' offloaded computation workloads, Wu et al. designed a non-orthogonal multiple access (NOMA) enabled MECO scheme, in which a group of users partially offload their computation tasks to a MECS via the NOMA-based transmission.

In multi-task single server offloading scenario, the authors in [22,23] proposed a three-step algorithm for jointly optimizing the resources allocation of both computation and communication in the

case with and without computing access point, respectively. Considering multiple users may usually offload their tasks to one MECS simultaneously, Chen et al. [24] used Lyaponuv Optimization Approach to determine the energy harvesting policy, and presented greedy maximal scheduling algorithms to solve the multi-task offloading problem for multiple users. In [25], the authors investigated the multi-task offloading in NOMA-based MEC, and adopted a two-step energy-efficient approach to obtain the total minimum energy consumption by optimizing computing resource allocation and the NOMA-transmission duration.

In addition, multi-user multi-task offloading in the case of multiple MECSs has been also attracted much attention. In this case, a critical issue is how to allocate edge computing resource to minimize the service cost and maximize the service capacity. To the end, Tran et al. [26] discussed the task offloading and resource allocation in such an environment and defined this problem as a mixed non-linear program to optimize MDs' transmission power, and computing resource allocation at the MECSs. The authors used quasi-convex and convex optimization schemes, and presented a heuristic algorithm as the solution. Dai et al. [15] proposed a two-tier computation offloading framework for multi-task in heterogeneous networks with multiple MECSs to minimize overall energy consumption. They jointly optimized user association and computation offloading while considering the computing resource allocation. Synthetically using local, edge and remote cloud computing models, the authors in [16] proposed a linear programing relaxation-based algorithm and a distributed deep learning-based offloading algorithm to guarantee QoS of the MEC network and to minimize MDs' energy consumption in the multi-user multi-task and multi-server MEC networks. Li et al. in [27] studied the MECO management problem in heterogenous network to minimize the network-level energy consumption and developed an iterative solution framework to obtain transmission power allocation strategy and computation offloading scheme.

The existing research work, although providing insights into diverse perspectives about computation offloading in MEC, has one common limitation: all of them assumed that the MECS could host all kinds of applications to compute various kinds of computation tasks. However, constrained by the MEC architecture, it is unrealistic to deploy as much computing resource as in cloud center on the MECSs, and each MECS can be only deployed limited types of applications. MECO problem in multi-task multi-server scenarios should take into account the collaboration of MECSs, and it is necessary to leverage the available resource on MECSs to design task offloading strategy. Therefore, the solutions proposed in previous works cannot be directly applicable to the problem in our work. In light of this, we elaborate in this paper a cross-sever multi-task offloading problem and design an efficient greedy algorithm to optimize the computing overhead in terms of accomplishing time and energy consumption.

## 3. System Model

In this paper, we mainly consider multi-task computation offloading scenario in multi-server MEC. As shown in Figure 1, $N$ access points (APs) $\mathcal{S} = \{1, 2, ..., N\}$ are located in a certain region, and each AP is installed with a MECS to provide enhanced computing service to the MDs. Please note that one AP may be a micro base station, or a small cell, or a WiFi access point. To easy present, in the following, the notation 'AP' refers to both AP and MECS. There are $M$ MDs, denoted as $\mathcal{U} = \{1, 2, ..., M\}$, randomly distributed in the coverage area of the $N$ APs. Suppose that there are $B$ orthogonal frequency channels denoted as $\mathcal{W} = \{1, 2, ..., B\}$ for each AP, and there exits an AP through which MD $i$ ($i \in \mathcal{U}$) can offload its computation tasks to the edge servers or to the remote cloud center (RCC). There are $K$ types computation tasks requested by all MDs, defined as $\mathcal{T} = \{t_1, t_2, ..., t_K\}$. Let $\mathcal{T}_i = \{t_{i1}, t_{i2}, ..., t_{iL}\}$ denote the set of tasks requested by MD $i$, where $\mathcal{T}_i \subset \mathcal{T}$, $L \leq K$. For MD $i$'s type-$j$ computation task, i.e., $t_{ij}$, it can generally be defined as $t_{ij} = (\alpha_{ij}, \beta_{ij}, \tau_{ij}^{max})$, where $\alpha_{ij}$ denotes the input data size of $t_{ij}$, $\beta_{ij}$ is the requisite CPU cycles for computing $t_{ij}$, and $\tau_{ij}^{max}$ is the maximum tolerated latency to accomplish $t_{ij}$. In particular, the information of $\beta_{ij}$ can be obtained by applying the methods proposed

in [28]. For convenience, the detailed notations and definitions used in this paper are summarized in Table 1.

To handle the computation tasks requested by MDs, corresponding applications must be deployed on the APs. The application hosting in the AP that can process type-$j$ task is called the APP-$j$. It is assumed that there exist $K$ types applications provided by the APs, denoted by $\mathcal{A} = \{a_1, a_2, ..., a_K\}$. Each AP $v$, with the maximum computing capacity $F_v$, can only host at most $Q$ types applications $\mathcal{A}_v = \{a_{v1}, a_{v2}, ..., a_{vQ}\}$, where $\mathcal{A}_v \subset \mathcal{A}$, $Q \leq K$. It is to say, one AP cannot handle all types of computation tasks, if AP $v$ does not host the corresponding APP-$j$, it must forward the task to other APs for processing, thus cross-server offloading is caused. As depicted in Figure 1, AP2 can directly compute task1 and task2 of MD1, while it must forward the tasks of MD2 to AP5, or to AP3 and AP4, respectively.

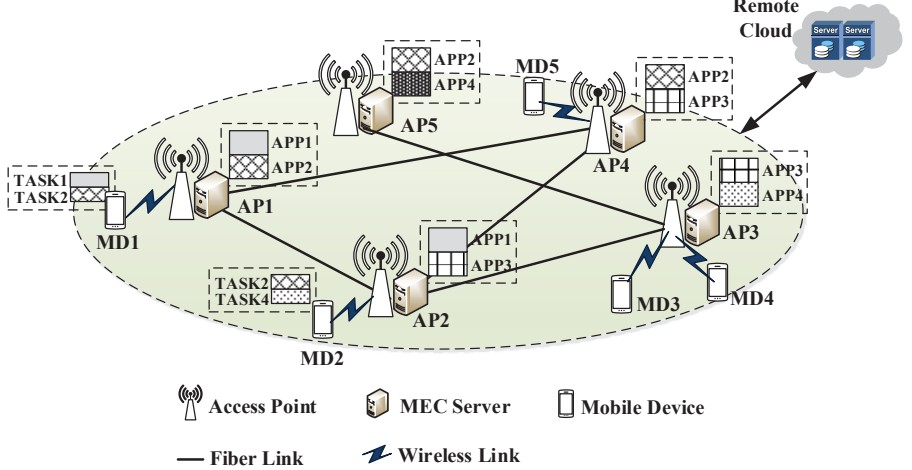

**Figure 1.** A multi-task multi-server MEC architecture: MEC servers deploying at the access points can process various types of tasks simultaneously. MDs can offload the tasks to MECSs or RCC.

**Table 1.** Notations and definitions

| Notation | Definition | Notation | Definition |
|---|---|---|---|
| $\mathcal{S}$ | set of APs and MECSs | $\mathcal{W}$ | set of wireless channels of each AP |
| $\mathcal{A}$ | APP types provided by all APs | $\mathcal{A}_v$ | APP types hosted on the AP $v$ |
| $\mathcal{U}$ | set of MDs | $\mathcal{T}$ | task types requested by all MDs |
| $\mathcal{T}_i$ | task types of MD $i$ | $t_{ij}$ | type-$j$ task of MD $i$ |
| $\alpha_{ij}$ | input data size of $t_{ij}$ | $\beta_{ij}$ | required CPU cycles to compute $t_{ij}$ |
| $\tau_{ij}^{max}$ | maximum latency to accomplish $t_{ij}$ | $\mu_{i,v}^b$ | selection of channel $b$ for MD $i$ |
| $\mu_{i,v}$ | channel selection of AP $v$ for MD $i$ | $w_b$ | bandwidth of channel $b$ |
| $\sigma$ | background noise power | $g_{i,v}^b$ | channel gain between MD $i$ and AP $v$ |
| $p_i$ | transmission power of MD $i$ | $I_{i,v}^b$ | interference of channel $b$ in AP $v$ |
| $r_{i,v}^b$ | data rate of MD $i$ accessing AP $v$ | $\rho_{ij}$ | task $t_{ij}$'s computing decision |
| $\tau_{ij}^{lc}$ | completing time in local computing | $e_{ij}^{lc}$ | consumed energy in local computing |
| $f_i^{lc}$ | computation capability of MD $i$ | $\eta_i^{lc}$ | consumed energy coefficient of MD $i$ |
| $\tau_{ij}^{ec}$ | completing time in edge computing | $e_{ij}^{ec}$ | consumed energy in edge computing |
| $\tau_{ij,tr}^{ec}$ | transmission delay of $t_{ij}$ | $\tau_{ij,fw}^{ec}$ | forwarding delay of $t_{ij}$ among APs |
| $\tau_{ij,co}^{ec}$ | computing delay of $t_{ij}$ on MECS | $\gamma_{ij,v}$ | AP $v$ hosting type-$j$ APP or not |
| $F_v$ | maximum computing capacity of AP $v$ | $f_{ij,v}^{ec}$ | computing ability allocated to $t_{ij}$ |
| $\tau_{ij}^{rc}$ | completing time in RCC | $e_{ij}^{rc}$ | consumed energy in RCC |
| $\tau_{ij,pr}^{rc}$ | propagation delay over fiber link | $\lambda_{ij,fw}$ | number of forwarding hops |

### 3.1. Communication Model

It is assumed that the MDs do not change their positions during the process of data transmission, and that each MD can only directly connect to one AP. Let the decision variable $\mu_{i,v}^b \in \{0,1\}$ denote channel selection of MD $i$. If MD $i$ chooses channel $b$ of AP $v$ to transmit the tasks, $\mu_{i,v}^b = 1$, otherwise, $\mu_{i,v}^b = 0$. Furthermore, we have $\mu_{i,v} = \sum_b \mu_{i,v}^b = 1$, if MD $i$ select AP $v$ to upload its tasks, and otherwise $\mu_{i,v} = \sum_b \mu_{i,v}^b = 0$. When MD $i$ access its closest AP $v$ via channel $b$, the data rate can be given as

$$r_{i,v}^b = w_b \log_2(1 + \frac{p_i g_{i,v}^b}{\sigma + I_{i,v}^b}), \tag{1}$$

where $w_b$ and $\sigma$ are separately the channel bandwidth and the background noise power, $p_i$ and $g_{i,v}^b$ are the transmission power of MD $i$ and the wireless channel gain, respectively. $I_{i,v}^b$ is the interference of channel $b$ suffered from other MDs using the same channel, which is calculated as

$$I_{i,v}^b = \sum_{l=1,l \neq v}^N \sum_{k=1,k \neq i}^M p_k \cdot g_{k,v}^b \cdot \mu_{k,l}^b. \tag{2}$$

### 3.2. Task Computing Model

Generally speaking, each computation task has totally three computing models to select, i.e., computing locally at its MD's CPU, processing on the edge servers, and offloading to the RCC. To optimize its computing overhead in terms of processing delay and energy consumption, MD $i$ can optimally select one from the three computing models for the type-$j$ task. In particular, we use $\rho_{ij} \in \{-1,0,1\}$ to denote task $t_{ij}$'s computing decision, where $\rho_{ij} = -1$ represents that MD $i$ decides to compute type-$j$ task locally, $\rho_{ij} = 0$ means that task $t_{ij}$ will be uploaded to the APs and be executed there, and $\rho_{ij} = 1$ is that MD $i$ chooses to compute its task $t_{ij}$ on the RCC.

*(1) Local Computing:*

When MD $i$ select local computing to accomplish type-$j$ task, i.e., $\rho_{ij} = -1$, the accomplishing time $\tau_{ij}^{lc}$ and the consumed energy $e_{ij}^{lc}$ of task $t_{ij}$ are calculated as

$$\tau_{ij}^{lc} = \frac{\beta_{ij}}{f_i^{lc}}, \tag{3}$$

$$e_{ij}^{lc} = \beta_{ij} \cdot \eta_i^{lc}. \tag{4}$$

In above equations, $f_i^{lc}$ and $\eta_i^{lc}$ denote the computation capability (i.e., CPU cycles per second) and the energy consumption coefficient for per CPU cycle of MD $i$, respectively.

*(2) Edge Computing:*

For the computing decision $\rho_{ij} = 0$, task $t_{ij}$ is chosen to be offloaded to the edge servers, and then will be computed on the servers. In this case, tasks requested by MDs will first be transmitted to their closest AP via the wireless link. Then the AP will compute the task if it has the corresponding applications. Otherwise, the tasks will be forwarded to other APs who host such applications through the fiber links among them, and at last be process there. Thus, the accomplishing time of task $t_{ij}$, $\tau_{ij}^{ec}$, consists of three parts: (i) $\tau_{ij,tr}^{ec} = \frac{\beta_{ij}}{r_{i,v}^b}$, the time cost for transmitting $t_{ij}$ via wireless channel between MD $i$ and the AP in close proximity, (ii) $\tau_{ij,fw}^{ec}$, the forwarding delay among APs, and (iii) $\tau_{ij,co}^{ec}$, the computing delay in corresponding AP. Here, we neglect delay for transmitting computed results from APs to MDs, since for most mobile applications, the data size of computed results is much smaller than that

of original input data. Let $\gamma_{j,v} = 1$ denote that AP $v$ hosts type-$j$ APP, and $\gamma_{j,v} = 0$ otherwise, then $\tau_{ij}^{ec}$ can be calculated as

$$\tau_{ij}^{ec} = \tau_{ij,tr}^{ec} + \gamma_{j,v} \cdot \tau_{j,co}^{ec} + (1 - \gamma_{j,v}) \cdot \tau_{ij,fw}^{ec}. \tag{5}$$

And the consumed energy can be given as

$$e_{ij}^{ec} = p_i \cdot \tau_{ij}^{ec}. \tag{6}$$

After transmitted to the AP $v$, task $t_{ij}$ will be processed if there exists the corresponding application. Let $f_{ij,v}^{ec}$ denote the computing ability of the AP $v$ allocated to task $t_{ij}$, the time for AP to process $t_{ij}$ is $\tau_{ij,co}^{ec} = \frac{\beta_{ij}}{f_{ij,v}^{ec}}$. If task $t_{ij}$ cannot find its corresponding application on the AP $v$, it must be routed to a target AP $d$ which hosts the required application. If we neglect the queueing time in each AP, the forwarding delay $\tau_{ij,fw}^{ec}$ can be given as

$$\tau_{ij,fw}^{ec} = \lambda_{ij,fw} \cdot \frac{m_{ij}}{c}, \tag{7}$$

where $\lambda_{ij,fw}$ is the number of routing hops between AP $v$ and $d$, and $c$ is the data transmission rate over fiber link.

*(3) Remote Computing:*

For the offloading decision $\rho_{ij} = 1$, MD $i$ will upload its computation task $t_{ij}$ to the RCC via the fiber backbone networks, and then the RCC server will compute the task. Notice that there are always rich computation resources deployed on the RCC, thus the computing time of $t_{ij}$ in this model can be neglected, and only the delay for $t_{ij}$ uploading via radio access and the propagation delay for transmitting the task through fiber link from the AP to the RCC are taken into account, i.e.,

$$\tau_{ij}^{rc} = \frac{\beta_{ij}}{r_{i,v}^b} + \tau_{ij,pr}^{rc}, \tag{8}$$

where $\tau_{ij,pr}^{rc}$ is the propagation delay. The consumed energy in this case is then given as

$$e_{ij}^{rc} = p_i \cdot \tau_{ij}^{rc}. \tag{9}$$

## 4. Cross-Server Multi-Task Computation Offloading

In this section, we first elaborate the problem formulation of cross-server multi-task computation offloading, and then propose our solutions to the problem.

### 4.1. Problem Formulation

From the analysis in Section 3, we can see that each task $t_{ij}$ can be either computed locally by using MD $i$'s CPU, or be processed on the APs, or be uploaded to the RCC for processing. According to different computing and offloading models, the accomplish time $\tau_{ij}$ and the consumed energy $e_{ij}$ of $t_{ij}$ may be diverse. Please note that most MDs are capacity-limited and battery-powered, our target is to make full use of computation resource provided by distributed APs and centralized RCC, and to devise an optimal computation offloading strategy so as to optimize the energy consumption, under the constraint of maximum tolerated latency for all MDs' computation tasks. Specifically, the problem of cross-server multi-task computation offloading can be formulated as follows.

**Definition 1.** *Cross-server Multi-task Computation Offloading (CMCO): Given the initial network information, including computation tasks requested by MDs, applications deployment on APs, the computing capability and the energy consumption per CPU cycle of MDs and APs, the transmit power of each MD, etc., the CMCO problem is to find an optimal computation offloading strategy for all MDs that minimizes the overall energy consumption with the constrained maximum tolerated accomplish delay of each task, while collaboratively using the computation resources of the MDs, the edge servers and the RCC.*

Based on the system model described in Section 3, the CMCO problem can be formulated as

$$\min_{\{\rho_{ij}\}} \sum_{i=1}^{M} \sum_{j=1}^{L} \rho_{ij} \cdot e_{ij}$$

$$\text{s.t.} \quad C1 : \rho_{ij} \cdot \tau_{ij} \leq \tau_{ij}^{max}, 1 \leq i \leq M, 1 \leq j \leq L,$$

$$C2 : \sum_{i=1}^{M} |\mu_{i,v}| \leq B, \forall s \in \mathcal{S},$$

$$C3 : \sum_{i=1}^{M} \sum_{j=1}^{N} \gamma_{j,v} \cdot f_{ij,v}^{ec} \leq F_v, \forall v \in \mathcal{S},$$

$$C4 : \rho_{ij} \in \{0, -1, 1\}, 1 \leq i \leq M, 1 \leq j \leq L,$$

$$C5 : \mu_{i,v} \in \{0, 1\}, 1 \leq i \leq M, \forall v \in \mathcal{S},$$

$$C6 : \gamma_{j,v} \in \{0, 1\}, 1 \leq j \leq L, \forall v \in \mathcal{S} \tag{10}$$

where $e_{ij}$ (i.e., the consumed energy of task $t_{ij}$ ) can be represented as

$$e_{ij} = \begin{cases} e_{ij}^{lc}, & if \quad \rho_{ij} = -1, \\ e_{ij}^{ec}, & if \quad \rho_{ij} = \quad 0, \\ e_{ij}^{rc}, & if \quad \rho_{ij} = \quad 1. \end{cases} \tag{11}$$

In (10), the constraints C1 guarantee the accomplishing time of $t_{ij}$ must be less than the maximum permissible latency, and C2 ensure that the totally used wireless channels of each AP by MDs cannot be exceed $B$. The constraints C3 guarantee that all computation resource allocated to tasks processed on each AP should not surpass its maximum computing capacity. The constraints C4 mean that each task $t_{ij}$ can choose only one offloading decision, C5 state that each MD can only directly communicate with one AP, and C6 declare whether the tasks can be processed on the given AP. Furthermore, it is easy to prove that the CMCO problem is NP-hard [29].

*4.2. Solutions*

*(1) An Basic Exhaustive Algorithm:* To solve the problem of CMCO, the most intuitive solution is to list all possible computation offloading decisions of all tasks, and then select the optimal one which can give the minimum overall consumed energy and meet the requirement of given permissible latency. Algorithm 1 describes the details of BEA.

---

**Algorithm 1** a basic exhaustive algorithm (BEA).

---

**Input:** $\mathcal{U}, \mathcal{S}, \mathcal{W}, \mathcal{T}_i, \mathcal{A}_v, p_i, f_i^{lc}, f_v^{ec}, \eta_i^{lc}, c, \sigma, \forall i \in \mathcal{U}, \forall v \in \mathcal{S}$.
**Output:** $\mathcal{D}$-the set of optimal offloading decision set, $E_{min}$-the minimum overall energy consumption.
1: **Initialize** $\mathcal{D} = \phi, E_{min} = +\infty$;
2: list all offloading decision combinations for each task, and compute their corresponding overall consumed energy;
3: desert those decision combinations which cannot satisfy the accomplishing time constraints of all tasks;
4: find the minimal overall energy consumption $E_{min}$ of the remained combinations, and record the corresponding combination into $\mathcal{D}$;
5: **return** $\mathcal{D}, E_{min}$.

---

Obviously, BEA described in Algorithm 1 is very effective, and it can surely find an optimal offloading decision to the CMCO problem. However, it must take an extremely long time to run, since BEA must list all optional combinations of offloading decision for all tasks. Because each task has three offloading decisions to select, it is easy to prove that the running time of BEA is $O(3^n)$, where $n$ is the total number of all tasks. Due to the long running time, BEA can only run in the MEC network where the number of tasks is very small, and it is unworkable in practice when the task number is very large. Here, BEA is presented just as a benchmark for the performance comparison of our GAA solution.

*(2) A Greedy Approximation Algorithm:* As BEA is impracticable for many computation tasks, we further design a greedy algorithm (GAA) to the CMCO problem. In GAA, we first call Procedure 1 to compute the accomplishing time ($\tau_{ij}$) and energy consumption ($e_{ij}$) of each task, and use set $\mathcal{C}_t$ to record the tasks whose $\tau_{ij}$ can meet their maximum tolerated latency. Then, we divide the tasks in $\mathcal{C}_t$ into three different sets $\mathcal{C}_{lc}, \mathcal{C}_{ec}$ and $\mathcal{C}_{rc}$, according to their $\tau_{ij}$ and $e_{ij}$ separately by adopting three different computing models. Next, the number of occupied wireless channels of the corresponding AP $s$, i.e., $ch\_num_s$, is iteratively updated. Finally, the total energy consumption and the corresponding optimal offloading decisions are obtained. The details of GAA are given in Algorithm 2.

---

**Algorithm 2** a greedy approximation algorithm (GAA).

---

**Input:** $\mathcal{U}, \mathcal{S}, \mathcal{W}, \mathcal{T}_i, \mathcal{A}_v, p_i, f_i^{lc}, f_v^{ec}, \eta_i^{lc}, c, \sigma, \forall i \in \mathcal{U}, \forall v \in \mathcal{S}$.
**Output:** $\mathcal{D}$-the set of optimal offloading decision set, $E_{min}$-the minimum overall energy consumption.
1: **Initialize** $ch\_num = \{0\}, \gamma_{ij} = \{0\}, \mathcal{C}_t = \mathcal{C}_{lc} = \mathcal{C}_{ec} = \mathcal{C}_{lc} = \phi$;
2: **for** each task $t_{ij}$ **do**
3:     call Procedure 1 to compute $\tau_{ij}^{ec}, e_{ij}^{ec}, \tau_{ij}^{lc}, e_{ij}^{lc}, \tau_{ij}^{rc}$, and $e_{ij}^{rc}$;
4:     record $t_{ij}$ into $\mathcal{C}_t$ if $\max\{\tau_{ij}^{lc}, \tau_{ij}^{ec}, \tau_{ij}^{rc}\} \leq \tau_{ij}^{max}$;
5: **end for**
6: **for** all $t_{ij} \in \mathcal{C}_t$ **do**
7:     **if** $(\tau_{ij}^{lc} > \tau_{ij}^{max}$ OR $\min\{e_{ij}^{ec}, e_{ij}^{rc}\} \leq e_{ij}^{lc})$ AND $ch\_num_v \leq B$ **then**
8:         record $t_{ij}$ into $\mathcal{C}_{ec}$ if $e_{ij}^{ec} < e_{ij}^{rc}$, otherwise, record $t_{ij}$ into $\mathcal{C}_{rc}$;
9:         $ch\_num_v = ch\_num_v + 1$;
10:     **else** record $t_{ij}$ into $\mathcal{C}_{lc}$;
11:     **end if**
12: **end for**
13: compute the overall energy consumption, $E_{min}, \mathcal{D} = \{\mathcal{C}_{lc}, \mathcal{C}_{ec}, \mathcal{C}_{rc}\}$;
14: **return** $\mathcal{D}, E_{min}$.

---

In Procedure 1, we mainly use open shortest path first (OSPF) routing strategy [30] to find the AP who hosts the corresponding APP-*j*. When the target AP is found and its maximum computing capacity is satisfied, the task $t_{ij}$ will be computed on this AP, then the corresponding $\tau_{ij}^{ec}$ and $e_{ij}^{ec}$ can be calculated.

---

**Procedure 1** a cross-server offloading procedure

---

1: $\lambda_{min} = +\infty$, $S_{id} = s$;
2: **for all** $d \in \mathcal{S}$ **do**
3:     search APP-$j$ on AP $d$;
4:     **if** MECS $d$ hosting APP-$j$ **then**
5:         calculate $\lambda_{v->d}$, the routing hops from AP $v$ to $d$ using OSPF routing strategy;
6:         $\gamma_{ij,d} = \gamma_{ij,d} + 1$;
7:     **end if**
8:     **if** $\lambda_{min} > \lambda_{v->d}$ AND $\gamma_{ij,d} \cdot f_{ij,d}^{ec} \leq F_d$ **then**
9:         $\lambda_{min} = \lambda_{v->d}$, $S_{id} = d$;
10:     **end if**
11: **end for**
12: compute $\tau_{ij}^{ec}$ and $e_{ij}^{ec}$;
13: compute $\tau_{ij}^{lc}$, $e_{ij}^{lc}$, $\tau_{ij}^{rc}$, and $e_{ij}^{rc}$;

---

As described in Algorithm 2, GAA uses a greedy method to select task $t_{ij}$'s computing model according the minimum consumed energy. Compared to BEA in Algorithm 1, although GAA can only achieve an approximation optimal result in the CMCO problem, it has a bigger advantage on computational efficiency, and can be applied in the MEC network with large number of tasks in practice. In particular, we have the following demonstration on the computational complexity of GAA.

**Proposition 1.** *GAA described in Algorithm 2 has the computational complexity of $O(n \cdot m^2)$, where m and n denote the MECS number and the total computation task number, respectively.*

**Proof of Proposition 1.** As described in Algorithm 2, GAA mainly includes three parts, i.e., recording the tasks satisfying their tolerated latency into $\mathcal{C}_t$, dividing these tasks into three different sets according to their accomplishing time and energy consumption, and computing the $E_{min}$. In particular, steps 2-5, i.e., the for loop, run $n$ times, and the running time of step 3, which calls Procedure 1 to compute $\tau_{ij}$, is $O(m^2)$ [30], where $m$ and $n$ are separately the number of APs and total computation tasks. Thus, the running time of steps 2–5 is $O(n \cdot m^2)$. From step 6 to 12, the for loop runs at most $n$ times, so steps 6-12 also need to run $O(n)$ time. Moreover, step 13 needs to run $n$ times to compute $E_{min}$. At last, the overall computational complexity of GAA in Algorithm 2 can be calculated as the sum of them, i.e., $O(n \cdot m^2)$. □

## 5. Numerical Results

In this section, we presented some performance evaluation to evaluate our proposed cross-server computation offloading schemes for multi-task mobile edge computing.

### 5.1. Experimental Settings

Without loss of generality, we took a MEC network with coexistence of RCC and MEC servers. There are 50 MDs distributed randomly in the coverage area of 10 APs, each of which has 50 orthogonal channels with the bandwidth 40 MHz. The neighboring APs are set 1 km apart from each other, and the distance from RCC to the APs is 1000 km. The data rate of the fiber link is assumed to be 1 Gbps. Each AP has the same computation capacity of 4 GHz, and the CPU frequency of a MD is 0.8~1 GHz. There are total 10 types of computation tasks and corresponding applications in the network, each MD has two types of tasks and each AP can host two types of APPs. Furthermore, each MD has the transmit power of 100 mW and the background noise is set as -100 dBm. The input data size of a task varies from 0.5~1 MB, and the CPU cycles to process a computation task $t_{ij}$ is assumed to be 0.1~1 Gigacycles. The maximum tolerated processing time for $t_{ij}$ is set as 0.1~2 s. All evaluations were performed on a PC with Intel i5-8265U CPU and 8.00GB RAM, using MATLAB R2016a in Windows 10 OS.

*5.2. Performance Evaluation*

The computation effectiveness of our proposed GAA algorithm are shown in Figures 2 and 3. To compare the performance between our solutions and existing algorithms, besides GAA and BEA, a well-known heuristics solution, simulated annealing algorithm (SAA) is also presented in our experiments, since SAA is widely adopted to solve NP-hard problem [31]. In addition, we use a randomized algorithm (RANA), which is repeated the random selection process 1000 times and taken the average value, to introduce statistical tests into our experiments.

Figure 2 presents the evaluation results among the three algorithms, in which x-axis is the number of MDs and y-axis denotes the computed minimum energy consumption. Considering the extremely high computational complexity, the number of MDs varies from 2 to 7 in the duration of this experiment. From Figure 2, we can see that GAA can obtain a near-optimal result in terms of minimum consumed energy compared to BEA and its performance outperform SAA when the number of MDS increasing, while RANA has a bad performance. It is confirmed by experimental results that GAA can achieve a near-optimal energy consumption to the CMCO problem. Figure 2 also presents that the consumed energy shows monotonic growth with the number of MDs increasing. Obviously, more MDs will provide more computation tasks to be processed, and more tasks will inevitably consume more energy.

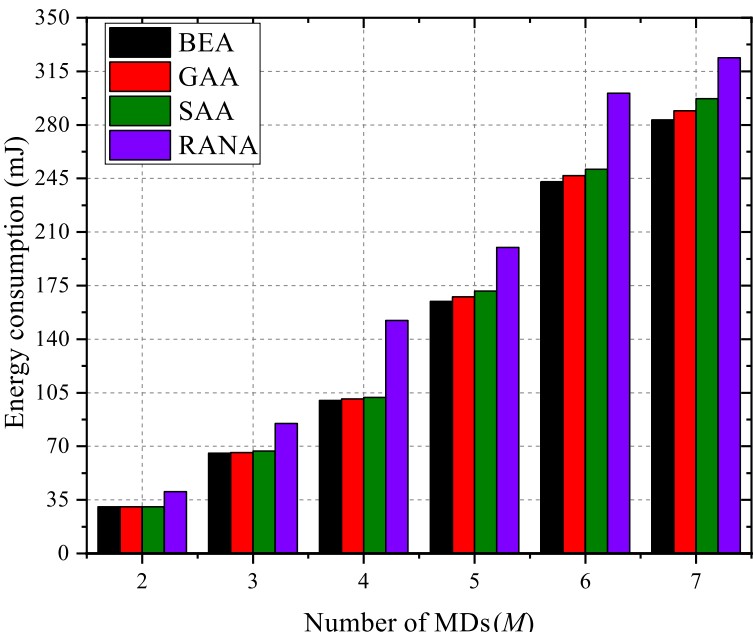

**Figure 2.** Performance comparison of the overall minimum energy consumption among BEA, GAA, SAA, and RANA with different numbers of MDs.

Figure 3 illustrates the computational complexity comparison among GAA, BEA, SAA, and RANA. From this figure, what can be clearly seen is that BEA has very high computational complexity than GAA and SAA, the running time of GAA is shorter than that of SAA. With the number of MDs increasing, the running time of BEA gets excessively larger while that of GAA remains almost unchangeable. Moreover, RANA has the lowest computational complexity. From above experimental evaluation, we can have the knowledge that although BEA gives the best solution to CMCO problem, it is not acceptable in practical applications due to its exceedingly high computational complexity. On the contrary, although GAA can only achieve near-optimal solutions, it has the advantage of very higher computing efficiency than BEA, and thus it could provide practicable solutions to the cross-server computation offloading in MEC.

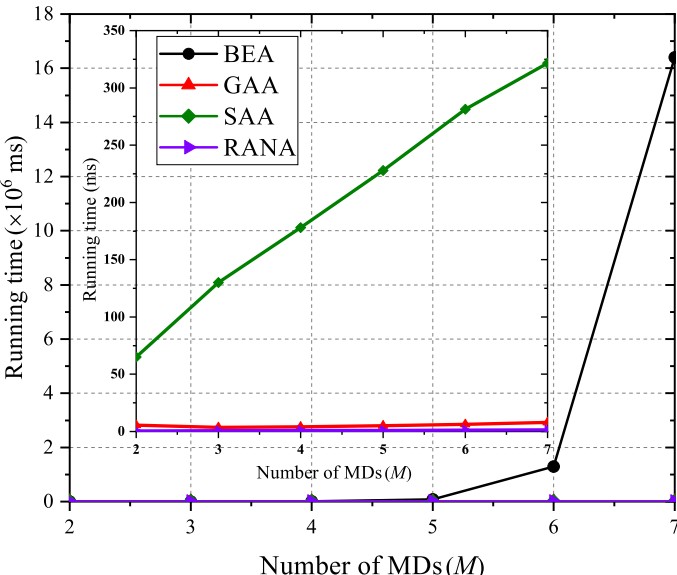

**Figure 3.** Illustration of the running time comparison among BEA, GAA, SAA, and RANA with different numbers of MDs.

To further confirm the advantages of our proposed GAA, we run experiments on GAA and SAA with more MDs, i.e., *M* varies from 10 to 100 with the increment of 10, and compare their obtained minimum energy consumption with those by adopting other computing models, i.e., computing locally by MDs' CPU, processing all the tasks on the MESs, and using GAA without RCC, respectively. Figure 4 presents the comparison results concerning these strategies. We can see from this figure, both GAA and SAA can obtain lower overall consumed energy than both local and edge computing models, which validate the necessity of task offloading. Moreover, one can also find that GAA always has better performance than SAA. Another aspect shown in Figure 4 is that the performance of GAA adopting all three computing models can achieve a little improvement over that of GAA without remote cloud offloading, which corroborates the superiority of collaborative computation offloading among the RCC, edge servers, and the MDs.

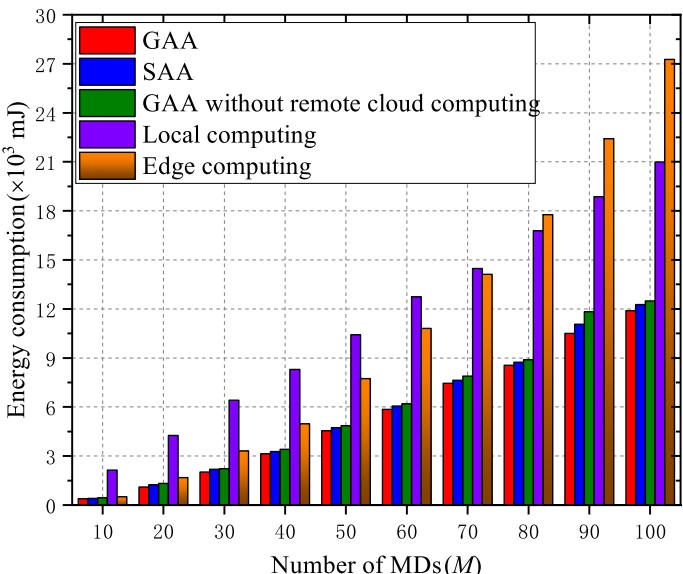

**Figure 4.** Illustration of the calculated minimum consumed energy results by adopting five different computing and offloading schemes, i.e., local computing, edge computing, GAA without remote offloading, and GAA as well as SAA methods.

From Figure 4, we can also know that when the number of MDS $M < 80$, the overall consumed energy of local computing is less than that of edge computing, while with the MDs number growing, the latter increases sharply and even exceeds the former. The reason is that with more and more tasks being uploaded to the APs via the wireless channels, the interference suffered from other MDs in the same channel increases, which decreases the uplink data rate, resulting in a greater computation overhead in terms of accomplishing time and energy consumption.

To identify how the computation resource of APs affects the total accomplishing time as well as the energy consumption, we deploy different number of APP types on each AP and compare the performances. This group of experiments is performed using GAA to get the near-optimal consumed energy. Here the number of MDs is set as 50 and that of APP types hosted on each MECS varies from 1 to 10. Figure 5 illustrates the comparisons of the obtained total energy consumption. From this figure, one can easily know that when the number of APP types, i.e., $Q$, increases, the total energy consumption decreases. When $Q \leq 9$, the obtained results reduces slowly, while when $Q = 10$, the total energy consumption shows a significant decline. This is due to the fact that if one AP hosts all types of APPs, it can process all types of computation tasks and need not forward them to other MECs, thus the total accomplishing time is reduced, and the energy consumption decreases.

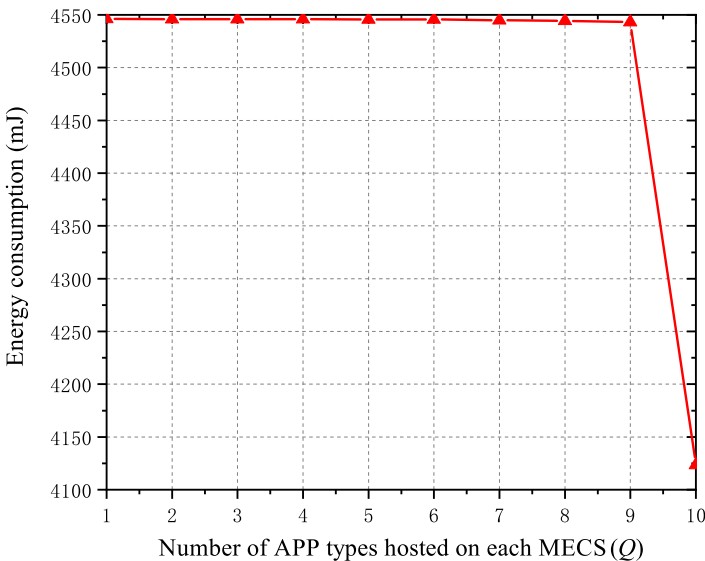

**Figure 5.** Comparisons illustration of the energy consumption with different number of APP types hosted on each AP.

### 5.3. Discussions

From above numerical results, we can observe that given the system model described in Section 3 and the corresponding parameter settings, GAA can efficiently solve the CMCO problem. However, the proposed system model and the formulated problem are all based on theoretical analysis, and the performance evaluation is performed using computer simulation, without experimental examination in real-world systems. Theoretical method can quickly construct system model and define the optimizing objective, but it usually needs some assumptions to neglect several network factors. While experimental techniques can precisely measure the actual system performance, this way may have strong pertinence and is lack of generality. Moreover, our system model mainly discusses the problem of cross-server task offloading and forwarding, which does not consider how to dispense one computational task to different edge servers in MEC, especially in heterogeneous distributed system [32].

## 6. Conclusions

In this paper, we have elaborated the problem of cross-server multi-task computation task offloading for energy consumption minimization in MEC networks. We first formulated this issue as a constrained optimization problem and then proposed a greedy approximation algorithm, i.e., GAA, as its solution. Different numbers of MDs and APP types, as well as various computing models were adopted in our performance evaluation. Experimental results validated that for the cross-server task offloading problem, GAA could achieve near-optimal performance compared to exhaustive algorithms with much shorter running time.

**Author Contributions:** Y.S. and Y.X. proposed the architecture of cross-server computation offloading MEC architecture. Y.S. and Y.G. proposed and analyzed the GAA algorithm. Y.S. and Y.X. performed the experiments. Y.S., Y.X. and Y.G. wrote the paper. All authors have read and agreed to the published version of the manuscript.

**Funding:** This work was funded by the Key Scientific Research Program of Higher Education (No. 20A510008 and No. 19A510018), the Foundation for Young Backbone Teachers in Higher Education Institutions (No. 2018GGJS126), and the Key Scientific and Technological Projects (No. 202102210120 and No. 192102210247), Henan Province, China.

**Conflicts of Interest:** The authors declare no conflict of interest.

## Abbreviations

The following abbreviations are used in this manuscript:

| | |
|---|---|
| AP | access point |
| APP | application |
| BEA | basic exhaustive algorithm |
| CMBP | classical maximum cardinality bin packing |
| CMCO | cross-server multi-task computation offloading |
| GAA | greedy approximation algorithm |
| IoT | Internet of Things |
| MD | Mobile device |
| MEC | mobile edge computing |
| MECO | MEC offloading |
| MECS | MEC server |
| NOMA | non-orthogonal multiple access |
| OSPF | open shortest path first |
| QoS | quality of service |
| RCC | remote cloud center |
| SAA | simulated annealing algorithm |

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
