# Peer review of "Cross-Server Computation Offloading for Multi-Task Mobile Edge Computing"

_information, doi:10.3390/info11020096_

Round 1

Reviewer 1 Report

The paper is well written and in general, easy to read. It presents a solution to an important problem in the context of edge computing development. 

There is however an important issue that needs to be taken into account and a few recommendations that should be followed to improve the work.

The important issue is the comparison with similar works to identify the shortcomings that need to be solved by the algorithm presented in this paper. It has to be noted that precisely the two works that seem to be more similar to the work presented in this paper (references [15] and [16]) are only mentioned in the introduction and not mention at all is done in the related work section. Reference [16] seems to follow an approach that is similar enough to the work presented in this paper as to be worth the comparison between these two works. 

The two things that are recommendable to improve the paper are the following:

All the problem formulation should be summarized all together on one page of the paper. I mean it is fine to have it through section 3 to explain the different models but, additionally, there should be one page with all the definitions. A page that could be easily and quickly read to have a clear understanding of the solution proposed. For instance:

          S = {1,2, ..., N}, N access points (APs) or MECS.

          A = {a1, a2, ... ak}, k types of tasks

          C- Maximum capacity of AP s

          As = {a1s, a2s, ... aQs}, Q types of tasks in AP s

          etc...

  I think that it is important to apply statistical tests to assure that the differences among the different options analyzed in the evaluation section are statistically meaningful. See for instance the paper https://doi.org/10.1002/stvr.1486 (A Hitchhiker's guide to statistical tests for assessing randomized algorithms in software engineering)

Other comments:

- I would not change from AP to MECS throughout the paper. Sometimes is confusing because the two terms are used in the same sentence. Even if authors indicate in the paper that they are interchangeable I would choose one of them and use always the same term along with the paper.

Author Response

The authors appreciate very much for your constructive comments on our manuscript, “Cross-server Computation Offloading for Multi-task Mobile Edge Computing” (Paper ID: information-697108).

We have revised our manuscript according to your comments. The revision details are summarized as follows.

Point 1: The important issue is the comparison with similar works to identify the shortcomings that need to be solved by the algorithm presented in this paper. It has to be noted that precisely the two works that seem to be more similar to the work presented in this paper (references [15] and [16]) are only mentioned in the introduction and not mention at all is done in the related work section. Reference [16] seems to follow an approach that is similar enough to the work presented in this paper as to be worth the comparison between these two works. 

Response 1: Thank you for your good comment. Just as you pointed out that, references [15] and [16] seem to be more similar to our work and not mentioned at all in Section II.  Reference [16] also studied the problem of multi-user multi-task and multi-server computation offloading in MEC networks. However, these two works assumed that the MEC servers could process all types of tasks requested by the MDs, neglecting the fact that only limited types of applications could be deployed on each MEC server. Our work aims to leverage the available resource on MEC servers to design task offloading strategy while taking into account the collaboration of servers. Therefore, the solutions proposed in previous works cannot be directly applicable to the problem in our work.

To address your comment, we have revised our manuscript and the following statements (Sections 2, Page 3) have been included in the revision.

Dai et al. [15] proposed a two-tier computation offloading framework for multi-task in heterogeneous networks with multiple MECSs to minimize overall energy consumption.  They jointly optimized user association and computation offloading while considering the computing resource allocation. Syntheticly utilizing local, edge and remote computing models, the authors in [16] proposed a linear programing relaxation-based algorithm and a distributed deep learning-based offloading algorithm to guarantee QoS of the MEC network and to minimize MDs’ energy consumption in the multi-user multi-task and multi-server MEC networks.

The existing research work, although providing insights into diverse perspectives about computation offloading in MEC, has one common limitation: all of them assumed that the MECS could host all kinds of applications to compute various kinds of computation tasks. However, constrained by the MEC architecture, it is unrealistic to deploy as much computing resource as in cloud center on the MECSs, and each MECS can be only deployed limited types of applications. MECO problem in multi-task multi-server scenarios should take into account the collaboration of MECSs, and it is necessary to leverage the available resource on MECSs to design task offloading strategy. Therefore, the solutions proposed in previous works cannot be directly applicable to the problem in our work.

Please kindly refer to Section 2 on Page 3 for revision.

Point 2: The two things that are recommendable to improve the paper are the following:

Point 2.1: All the problem formulation should be summarized all together on one page of the paper. I mean it is fine to have it through section 3 to explain the different models but, additionally, there should be one page with all the definitions. A page that could be easily and quickly read to have a clear understanding of the solution proposed. For instance:

          S = {1,2, ..., N}, N access points (APs) or MECS.

          A = {a1, a2, ... ak}, k types of tasks

          Cs - Maximum capacity of AP s

          As = {a1s, a2s, ... aQs}, Q types of tasks in AP s

          etc...

Response 2.1: Thank you for your god comment. To address your comment, we have reorganized Section III to present the system model more clearly. In addition, we also add a table (Table 1) to list all notations and definitions used in the paper, so as to make our paper more readable.

       Please kindly refer to Section 3 on Pages 3-4 for revision.

Point 2.2: I think that it is important to apply statistical tests to assure that the differences among the different options analyzed in the evaluation section are statistically meaningful. See for instance the paper https://doi.org/10.1002/stvr.1486 (A Hitchhiker's guide to statistical tests for assessing randomized algorithms in software engineering)

Response 2.2: Thank you for your god comment. To address your comment, we have added a new algorithm based randomized method, named as RANA, in the performance evaluation section. In addition, the following statements (Section 5.2 on Page 9) have been also included in the revision.

In addition, we utilize a randomized algorithm (RANA), which is repeated the random selection process 1000 times and taken the average value, to introduce statistical tests into  our experiments.

     Please kindly refer to Section 5.2 on Pages 9-10 for revision.

Point 3: I would not change from AP to MECS throughout the paper. Sometimes is confusing because the two terms are used in the same sentence. Even if authors indicate in the paper that they are interchangeable I would choose one of them and use always the same term along with the paper.

Response 3: This is a nice comment. To address your comment, we have revised our manuscript and used ‘AP’ to represent both AP and MECS along with the paper.

      Please kindly refer to the manuscript for revision.

Reviewer 2 Report

Essential comments:

The Authors deal with a problem of the overall consumed energy minimization in multi-server and multi-task mobile edge computation offloading taking into account local computing, edge computing and remote computing. They proposed a heuristic greedy approximation algorithm to solve this problem. Mathematical system model is also presented considering communication model, computation task model and task computing model. The proposed algorithm has been validated via computer simulation and compared to basic exhaustive algorithm and to simulated annealing algorithm.

Work presented in the manuscript is interesting. The topic is up-to-date, especially in the era of mobiles and 5G technology. Mathematical model of the system and original heuristic algorithm are strong points of this submission. However the paper has some shortcomings, which should be improved before the publication.

1) Description of the system model is unclear and ambiguous.

There is a mess in variable markings and indexes. For example, in line 111 access points (APs) are marked by S (capital S), and in line 113 by s (small s) without index, when in line145 by s and d. In line 118 for maximum computing capacity Cs, s is upper index, for applications As, s is index, and for single application as1, ..., asi, s is upper index again. This is confusing. Strict and clear notation and its explanation is needed.

A list of markings is missing. This also makes description of the system model hard to follow, especially due to the large number of markings explained in the text not under equations. A list of abbreviations is placed in the end of the manuscript, however some markings and abbreviations, e.g.: OSPF, are not explained. All markings and abbreviations should be explained at their first use. Variables, subscripts and upper indices should be explained under equations. Names of variables should be unified in the whole paper.

2) The state of art and description of related works is extensive. However, the Authors should more precisely explain what differences are between their work and publication listed on position [16] in the reference list, and compare applied solutions.

They should also refer to the following publication:

Li, S.; Tao, Y.; Qin, X.; Liu, L.; Zhang, Z.; Zhang, P. Energy-Aware Mobile Edge Computation Offloading for IoT Over Heterogenous Networks. IEEE Access 2019, 7, 13092-13105.

Both papers deal with multi-server and multi-task mobile edge computation offloading and energy consumption optimisation considering local, edge and remote computing.

3) It is not clear, how values of variables and parameters of system model had been evaluated before the computer simulation. Did the Authors considered any experiments in a real distributed heterogenous system. Did they used any benchmark for evaluation of network performance for different considered cases? I would expect that the Authors will refer to this issue and comment advantages and disadvantages of theoretical and experimental methods. They should also refer to the literature dealing with evaluation of network performance and distributed system design, as for example to the position listed below:

Majchrowicz M., Kapusta P., Jackowska-Strumiłło L., Sankowski D.: Multi-GPU, multi-node algorithms for acceleration of image reconstruction in 3D Electrical Capacitance Tomography in heterogeneous distributed system. Sensors, 2020, 20, 391; doi:10.3390/s20020391

The Authors should also point out limitations of their system model used in simulations in relation to real distributed heterogenous systems.

4) There are some inconsistencies in description of Algorithm 2. Procedure 1 is called twice, what is redundant. In addition, the procedure returns different data each time when it is called, i.e: first it returns times, next energies. Are there two procedures, or maybe this is a script? Also, in Algorithm 2: if the condition in step 4 is true then the first relation (left argument of OR function) in step 8 is always false and it need not to be checked once again. This whole description is not professional.

Editorial comments:

In general, the paper is well written and well organized. I have noticed only small editorial mistakes.

1) In section 5.2, in line 231 the sentence “We first use Figure 2 and Figure 3 to verify the effectiveness of our proposed GAA algorithm.” is not well formed. Results of this verification are shown in the figures.

2) In lines 57-58 the Authors write about the enumeration algorithm (BEA), and in line 178 about basic exhaustive algorithm (BEA), while in line 67 is BEOA (not explained here), and in line 298 (Abbreviations ) is BEOA basic optimal enumeration algorithm. This should be ordered.

Author Response

Response to Reviewer 2 Comments

The authors appreciate very much for your constructive comments on our manuscript, “Cross-server Computation Offloading for Multi-task Mobile Edge Computing” (Paper ID: information-697108).

We have revised our manuscript according to your comments. The revision details are summarized as follows.

Point 1: Description of the system model is unclear and ambiguous.

There is a mess in variable markings and indexes. For example, in line 111 access points (APs) are marked by S (capital S), and in line 113 by s (small s) without index, when in line145 by s and d. In line 118 for maximum computing capacity Cs, s is upper index, for applications As, s is index, and for single application as1, ..., asi, s is upper index again. This is confusing. Strict and clear notation and its explanation is needed.

A list of markings is missing. This also makes description of the system model hard to follow, especially due to the large number of markings explained in the text not under equations. A list of abbreviations is placed in the end of the manuscript, however some markings and abbreviations, e.g.: OSPF, are not explained. All markings and abbreviations should be explained at their first use. Variables, subscripts and upper indices should be explained under equations. Names of variables should be unified in the whole paper.

Response 1: Thank you for your good comment. Just as you pointed out that, there is a mess in variable markings and indexes, and list of marking is missing in our previous manuscript. To address your comment, we have revised our manuscript and the following statements have been included in the revision.

  (1) We have reorganized Section III to present the system model more clearly, and redefined the notations using the uniform symbols and indexes. In addition, we also add a table (Table 1) to list all notations and definitions used in the paper, so as to make our paper more readable.

    Please kindly refer to Section 3 on Pages 3-4 for revision.

(2) We also revised our manuscript and given the all the abbreviations and their explanations used in this paper at the end of the manuscript.

 Please kindly refer to Abbreviations on Page 13 for revision.

Point 2: The state of art and description of related works is extensive. However, the Authors should more precisely explain what differences are between their work and publication listed on position [16] in the reference list, and compare applied solutions.

They should also refer to the following publication:

Li, S.; Tao, Y.; Qin, X.; Liu, L.; Zhang, Z.; Zhang, P. Energy-Aware Mobile Edge Computation Offloading for IoT Over Heterogenous Networks. IEEE Access 2019, 7, 13092-13105.

    Both papers deal with multi-server and multi-task mobile edge computation offloading and energy consumption optimisation considering local, edge and remote computing.

Response 2 : Thank you for your good comment. Just as you pointed out that, there are lack of explanations about reference [16], and the difference between the work in [16] and ours should be discussed more precisely.  Bothe references [16] and [27] studied the problem of multi-user multi-task and multi-server computation offloading in MEC networks considering local, edge and remote computing. However, these two works assumed that the MEC servers could process all types of tasks requested by the MDs, neglecting the fact that only limited types of applications could be deployed on each MEC server. Our work aims to leverage the available resource on MEC servers to design task offloading strategy while taking into account the collaboration of servers. Therefore, the solutions proposed in previous works cannot be directly applicable to the problem in our work.  

     To address your comment, we have revised our manuscript and the following statements (Sections I, Page 3) have been included in the revision.

Syntheticly utilizing local, edge and remote computing models, the authors in [16] proposed a linear programing relaxation-based algorithm and a distributed deep learning-based offloading algorithm to guarantee QoS of the MEC network and to minimize MDs’ energy consumption in the multi-user multi-task and multi-server MEC networks. While Li et al. in [27] studied the MECO management problem in heterogenous network to minimize the network-level energy consumption and developed an iterative solution framework to obtain transmission power allocation strategy and computation offloading scheme.

   “The existing research work, although providing insights into diverse perspectives about computation offloading in MEC, has one common limitation: all of them assumed that the MECS could host all kinds of applications to compute various kinds of computation tasks. However, constrained by the MEC architecture, it is unrealistic to deploy as much computing resource as in cloud center on the MECSs, and each MECS can be only deployed limited types of applications. MECO problem in multi-task multi-server scenarios should take into account the collaboration of MECSs, and it is necessary to leverage the available resource on MECSs to design task offloading strategy. Therefore, the solutions proposed in previous works cannot be directly applicable to the problem in our work.

    Also, the following new reference has been included in this revision to backup our discussion:

 27. Li, S.; Tao, Y.; Qin, X.; Liu, L.; Zhang, Z.; Zhang, P. Energy-Aware Mobile Edge Computation Offloading for IoT Over Heterogenous Networks. IEEE Access 2019, 7, 13092-13105.

 Please kindly refer to Section 2 on Page 3 for revision.

Point 3: It is not clear, how values of variables and parameters of system model had been evaluated before the computer simulation. Did the Authors considered any experiments in a real distributed heterogeneous system. Did they used any benchmark for evaluation of network performance for different considered cases? I would expect that the Authors will refer to this issue and comment advantages and disadvantages of theoretical and experimental methods. They should also refer to the literature dealing with evaluation of network performance and distributed system design, as for example to the position listed below:

Majchrowicz M., Kapusta P., Jackowska-Strumiłło L., Sankowski D.: Multi-GPU, multi-node algorithms for acceleration of image reconstruction in 3D Electrical Capacitance Tomography in heterogeneous distributed system. Sensors, 2020, 20, 391; doi:10.3390/s20020391

The Authors should also point out limitations of their system model used in simulations in relation to real distributed heterogeneous systems

Response 3: Thank you for your nice comment.

   Just as you understood that it is not clear how values of variables and parameters of system model had been evaluated before the computer simulation. We agree with you that the advantages and disadvantages of theoretical and experimental methods should be further commented in the paper.

  We would like to take this opportunity to state our main contribution is to discuss the cross-server multi-task computation offloading problem in MEC network. To the best of our knowledge, we are the first to study the collaborative offloading problem in multi-server in MEC since existing works concerning the task offloading problem mainly assumed that the MEC servers could process all types of tasks and neglected the limited applications deployed on MEC servers. Therefore, as a first step toward such a problem, we consider in this paper a relatively simple system model and propose a greedy algorithm to minimize the total consumed energy. To evaluate the network performance, we use the basic exhaustive algorithm as benchmark and adopt the parameter settings presented in available research works. What’s more, due to the limitation of our experimental conditions, we have not considered any experiments in a real distributed heterogeneous system.

   As you suggested that it would be very valuable to discuss the advantages and disadvantages of theoretical and experimental methods, and point out limitations of their system model used in simulations in relation to real distributed heterogeneous systems. To address your comment, we have revised our manuscript and the following statements (Section 5.3 on Page 12) have been included in the revision.

 From above numerical results, we can observe that, given the system model described in Section 3 and the corresponding parameter settings, GAA can efficiently solve the CMCO problem. However, the proposed system model and the formulated problem are all based on theoretical analysis, and the performance evaluation is performed using computer simulation, without experimental examination in real-world systems. Theoretical method is able to quickly construct system model and define the optimizing objective, but it usually needs some assumptions to neglect several network factors. While experimental techniques can precisely measure the actual system performance, yet this way may have strong pertinence and is lack of generality. Moreover, our system model mainly discusses the problem of cross-server task offloading and forwarding, which does not consider how to dispense one computational task to different edge servers in MEC, especially in heterogeneous distributed system [32].

Also, the following new reference has been included in this revision to backup our discussion:

32. Majchrowicz, M.; Kapusta, P.; Jackowska-StrumiÅ‚Å‚o, L.; Banasiak, R.; Sankowski, D. Multi-GPU, multi-node algorithms for acceleration of image reconstruction in 3D Electrical Capacitance Tomography in heterogeneous distributed system. Sensors 2020, 20, 1–20.

     Please kindly refer to Section 5.3 on Page 12 for revision.

Point 4: There are some inconsistencies in description of Algorithm 2. Procedure 1 is called twice, what is redundant. In addition, the procedure returns different data each time when it is called, i.e: first it returns times, next energies. Are there two procedures, or maybe this is a script? Also, in Algorithm 2: if the condition in step 4 is true then the first relation (left argument of OR function) in step 8 is always false and it need not to be checked once again. This whole description is not professional.

Response 4: Thank you for your nice comment. This confusion was caused by our improper writing.  In fact, Procedure 1 mainly cope with the problem of cross-server task forwarding. When Procedure 1 is called, it can return all the results simultaneously including accomplishing time and consumed energy in the three computing models.

     In Algorithm 2, step 4 is used to record the tasks into set Ct whose accomplishing time satisfy their maximum tolerated latency when adopting different computing models (i.e., ).  In this step, no matter what computing model is adopted, as long as the completing time meets the latency constraint, the corresponding task will be recorded into Ct. That is, this step cannot ensure  . Therefore, if the condition in step 4 is true, the first relation (left argument of OR function) in step 8, i.e., , may be true.

      To address your comment, we have rewritten GAA in Algorithm 2 and revised the Proof of Proposition 1. Please kindly refer to Section 4.2 on Pages 8-9 for revision.

Point 5: Editorial comments:

In section 5.2, in line 231 the sentence “We first use Figure 2 and Figure 3 to verify the effectiveness of our proposed GAA algorithm.” is not well formed. Results of this verification are shown in the figures.

In lines 57-58 the Authors write about the enumeration algorithm (BEA), and in line 178 about basic exhaustive algorithm (BEA), while in line 67 is BEOA (not explained here), and in line 298 (Abbreviations ) is BEOA basic optimal enumeration algorithm. This should be ordered.

Response 5: Thank you very much for your careful checking. We have checked the manuscript very carefully one more time to remove all the typos, so as to improve the paper presentation.

Please kindly refer to Section 1 on Page 2, Section 5.2 on Page 9, and Abbreviations on Page 13 for revision.

Round 2

Reviewer 1 Report

Thanks to the author for taking into account most of my comments in their review.

I only have one concern regarding one of my prior comments since I consider that the authors did not fully cope with it in the new version of the paper. 

My previous comment was: "I think that it is important to apply statistical tests to assure that the differences among the different options analyzed in the evaluation section are statistically meaningful. See for instance the paper https://doi.org/10.1002/stvr.1486 (A Hitchhiker's guide to statistical tests for assessing randomized algorithms in software engineering)"

Although some actions were taken in the new version of the paper and a new random algorithm was included in the evaluation this is not was I was referring to. My question is: are the difference obtained for the different algorithms under different circumstances statistically meaningful? 

Anyway, my impression is that it is feasible to leave this additional evaluation for future work. This means that in this paper, at least,  I would include a comment about this as part of the conclusions/future work section. 

Reviewer 2 Report

The Authors answered to all my comments and corrected the manuscript. I have no more comments.